# Explainable Deep Learning for Disease Activity Prediction in Chronic Inflammatory Joint Diseases

**Cécile Trottet** [1]  **Ahmed Allam** [1]  **Aron N. Horvath** [1]  **Raphael Micheroli** [2]  **Michael Krauthammer** [1 3 *]
**Caroline Ospelt** [2 *]

## Abstract

Analysing complex diseases such as chronic inflammatory joint diseases, where many factors influence the disease evolution, is a challenging task. We propose an explainable attention-based neural network model trained on data from patients with different arthritis subtypes for predicting future disease activity scores. The network transforms longitudinal patient journeys into comparable representations allowing for additional case-based explanations via computed patient journey similarities. We show how these similarities allow us to rank different patient characteristics in terms of impact on disease progression and discuss how case-based explanations can enhance the transparency of deep learning solutions.

## 1. Motivation

Disease progression patterns in chronic inflammatory joint diseases (CIJDs), as recorded in CIJD registries, are complex and patient-specific. As a result, these registries are very heterogeneous and irregular in both the temporal and the recorded features aspect (i.e. varying number of medical visits and recorded measurements). In this work, we propose an explainable multi-task model for transforming longitudinal patient data (patient journeys) from a Swiss CIJD registry into comparable representations and predicting future disease activity in CIJDs. Our model evaluates the importance of the different aspects of individual management history to predict CIJD progression using different approaches of model explainability.

---
*Equal contribution [1]Department of Quantitative Biomedicine, University of Zurich, Zurich, Switzerland [2]Department of Rheumatology, University Hospital Zurich, University of Zurich, Zurich, Switzerland [3]Biomedical Informatics DFL, University Hospital Zurich, University of Zurich, Zurich, Switzerland. Correspondence to: Cécile Trottet <cecileclaire.trottet@uzh.ch>.

*Workshop on Interpretable ML in Healthcare at International Conference on Machine Learning (ICML)*, Honolulu, Hawaii, USA. 2023.

To this end, we examined (1) *model-based* explainability by inspecting the attribution scores from the attention layers in our model's architecture, and (2) *case-based* explanations by designing a feature importance weighting method coupled with patient similarity assessment. By contrasting the results of both approaches, we believe that we make a significant step towards enhancing the transparency of the model's output.

### 1.1. Related work

There is limited research on employing temporal modelling approaches to model disease progression in CIJDs. In the existing studies, the continuous disease activity scores (DAS) are usually simplified and thresholded into a binary classification task such as remission/no remission or response/no response, rather than predicted through regression (Montani & Striani, 2019). For instance, Norgeot et al. (2019) implemented recurrent neural networks to predict disease activity (remission/no remission) at the next rheumatology visit, and Lee et al. (2021) implemented non-temporal ML models to predict response/no response to different treatments. Overall, their results support our findings that past measurements of disease activity are highly predictive of disease progression and that temporal models outperform static baselines. Our model architecture builds on the work of Kalweit et al. (2021), and further extends it with the addition of attention and multi-task layers to support patients with different CIJD subtypes allowing us to analyse the role of patient history on model predictions.

## 2. Materials and methods

### 2.1. Dataset

The Swiss Clinical Quality Management in Rheumatic Diseases Foundation (SCQM) (Uitz et al., 2000) maintains a national registry of inflammatory rheumatic diseases. The database documents the disease management over time for $19'267$ patients through clinical measurements (CM) during the visits, demographics (Dem), prescribed medications (Med) and patient-reported outcome measures (PROM).

## 2.2. Model architecture

We adapted the architecture proposed by Kalweit at al. (2021) to our setting by training multiple LSTMs (Hochreiter & Schmidhuber, 1997), multiple prediction networks, and by augmenting the model with several attention layers (Vaswani et al.) (Figure 1). We refer to the complete model by the DAS-Net (Disease Activity Score Network).

DAS-Net takes as input the patient's medications, clinical measurements, PROM up to a chosen time point, demographics (non-temporal) and the time to prediction. The different sources of information (i.e. events) in the patient histories are handled separately until aggregation occurs in the representation layers, as these measurements are not aligned in time and contain different features. DAS-Net predicts the future DAS28 or ASDAS[1] score by feeding the computed latent representation in the penultimate layers to two separate blocks of prediction layers.

## 2.3. Patient similarity ($k-$NN regression model)

We evaluated the utility of the computed DAS-Net's latent representations to retrieve similar patients. Given a patient representation at a prediction time-point, we computed the $L^1$ distance to all other representations and selected the $k$ closest patient embeddings ($k = 50$). Analogous to $k-$NN regression, we compared the representation's future DAS with the average DAS of their closest matched set.

### 2.3.1. FEATURE-LEVEL IMPORTANCE

Using the sets of similar patients computed by the $k-$NN model, we designed a feature importance ranking method to gain insights into patient feature-level importance.

We examined which features tended to have similar values within sets of similar patients. Specifically, we compared the distributions of the features within sets of similar patients to their distributions in the entire cohort. We then ranked the features by their likelihood of displaying similar values among sets of similar patients. The details of the computations for continuous and categorical features are in appendix A.1.

## 3. Results

We compared the performance of DAS-Net to a vanilla neural network (MLP), and tree-based gradient boosting model (XGBoost). We also assessed the performance of the $k$-NN regression model and further explored both explainability approaches to better understand the relationship between input features and model output at different stages of the modelling process.

---

[1] These two scores measure the disease activity in CIJDs.

## 3.1. Performance

DAS-Net achieves the lowest mean squared error on both prediction tasks (MSEs of $0.510 \pm 0.009$ for ASDAS and $0.965 \pm 0.014$ for DAS28) (Table 1) compared to the two baseline models (MLP, XGBoost). The $k$-NN model also outperforms the baseline models (Table 1) suggesting that DAS-Net latent representations successfully capture the important predictive components from the patient history.

## 3.2. Model-based explainability

Our model employs a two-layered attention mechanism for model-based explainability. It assigns weights to the different events of the patient histories, highlighting their significance for the predictions.

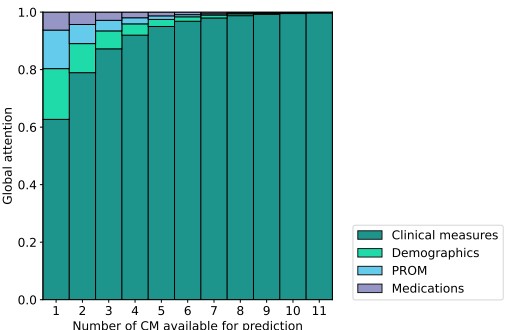

(a) **Average global attention** on the test set for the different events.

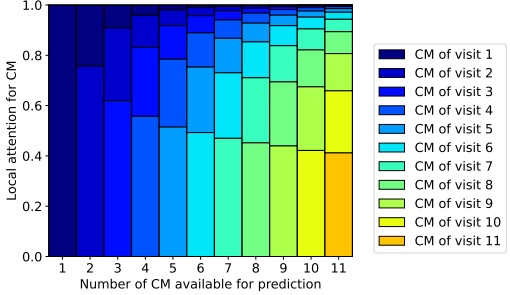

(b) **Average local attention** on the test set for clinical measures.

Figure 2: **Attention** weights for increasing history lengths.

### 3.2.1. GLOBAL ATTENTION

The **global attention** weighs the aggregated temporal histories and demographics when building the patient's full history representation. It shows which type of event (i.e. CM, Med, PROM or Dem) is weighted the most by the model when making predictions.

Figure 2a shows the attribution of the global attention weights to the different event features in the patients' history as the history length increases. When limited information

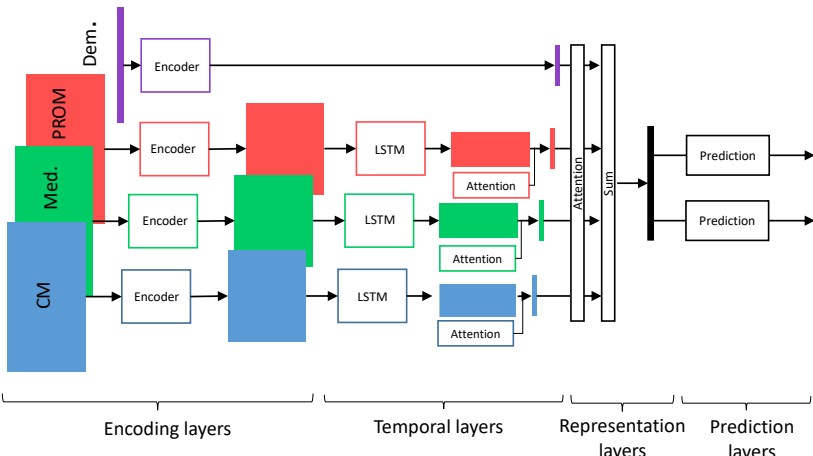

Figure 1: **Model architecture.** The encoders and predictions networks are MLPs. The model uses LSTMs to aggregate input sequences of different lengths and attention mechanism to weigh the different components of the input.

Table 1: **Model performance on test set for prediction of the two target DAS at the next medical visit.**

| Framework | Model | MSE ASDAS | MSE DAS28 |
|---|---|---|---|
| **Prediction** | DAS-Net | **0.510 ± 0.009** | **0.965 ± 0.014** |
| | XGBoost | 0.534 ± 0.003 | 0.992 ± 0.002 |
| | MLP | 0.562 ± 0.005 | 1.029 ± 0.007 |
| **Similarity** | $k$-NN model on DAS-Net latent representations | **0.506** | **0.966** |

is available, the model considers all the sources of information. With increasing history lengths, the model increasingly assigns higher weights to the past CM (clinical measures) compared to the other sources of information. This weighting occurs because the previous CM contain the previous DAS that is predictive of the future DAS.

### 3.2.2. LOCAL ATTENTION

The **local attention** is specific to each type of time-related event, showing the weight given to each instance (for example a specific medication) when building the patient representation. Since the highest global attention weights are attributed to the CM, we further inspected the attribution of the local attention weights for the CM in the patient's history when predicting the target outcome (DAS).

Figure 2b shows that most attention is directed at the last available CM in the history before the prediction. Our model thus assigns the highest attention scores to the recent CM, particularly the ones preceding the prediction.

### 3.3. Case-based explainability

We also used case-based similarity strategies for providing visual explanations and to determine the importance of individual patient features for disease progression prediction.

### 3.3.1. VISUALISATIONS

We plotted the two-dimensional t-SNE embeddings (Van Der Maaten & Hinton, 2008) of the patient latent representations. In Figure 3 we overlaid the embeddings with colourmaps reflecting the values of the features, reporting the last available feature value at a given time.

The plots provide general visual insight into the representation space. For instance, Figure 3d shows the repartition of the smoker statuses, where embeddings in the top left subspace correspond to patients with a smoking status that seems determinant for their DAS prediction.

Furthermore, in Figure 3 we highlighted an embedding $e_{p,t}$ from a patient $p$ at time $t$ (larger dot) and its nearest neighbours $\mathcal{N}_e$ as computed by the $k-$NN model (triangles). For continuous features, we report the average value in the entire representation set $\mathcal{R}$ and in $\mathcal{N}_e$. For categorical features, we report the incidence of each category in $\mathcal{R}$ and $\mathcal{N}_e$. By comparing the overall distribution of the feature with its distribution within $\mathcal{N}_e$, we get insight into its given importance for the similarity assessment. For the example patient in Figure 3, its smoker status (Figure 3d) and gender (Figure 3b) seem decisive for the similarity assessment since all of her nearest neighbours are also smoking females.

### 3.3.2. FEATURE RANKING BY SIMILARITY

Plots in Figure 3 provide insights into the nearest neighbour attribution mechanism for individual cases. Using the method described in 2.3.1 and A.1, we ranked the features by global importance in the cohort and found that both DAS scores and the number of painful joints are the most important for the similarity assessment for continuous features. Similarly, the high duration of morning stiffness and gender of patients are the top-2 categorical features. The rankings of the features by importance are listed in Tables 2 and 3 of the appendix and the formulas for the computation are reported in subsection A.1.

## 4. Conclusion

We propose DAS-Net, a modular recurrent neural network and attention-based model for predicting future disease activity in CIJDs that outperforms non-temporal baseline models. Model-based explainability shows that DAS-Net relies on recent information while still attributing significant weight to older events. Past disease activity scores were consistently the strongest predictors. We find that a $k$-NN-based approach driven by computing patient similarities from the model's latent representations has a similar prediction performance to DAS-Net, showing that our modelling approach is well suited to transforming heterogeneous medical records into comparable and meaningful representations. Using these representations, we can find sets of similar patients that allow us to derive global feature importance.

Overall, our study demonstrates promising results towards developing an explainable clinical decision support system for retrieving similar patients and predicting their disease progression while considering the different disease management strategies that worked best for similar patients.

## 5. Acknowledgements

The authors thank the patients and the rheumatologists who made the study possible, as well as the SCQM collaborators for data management.

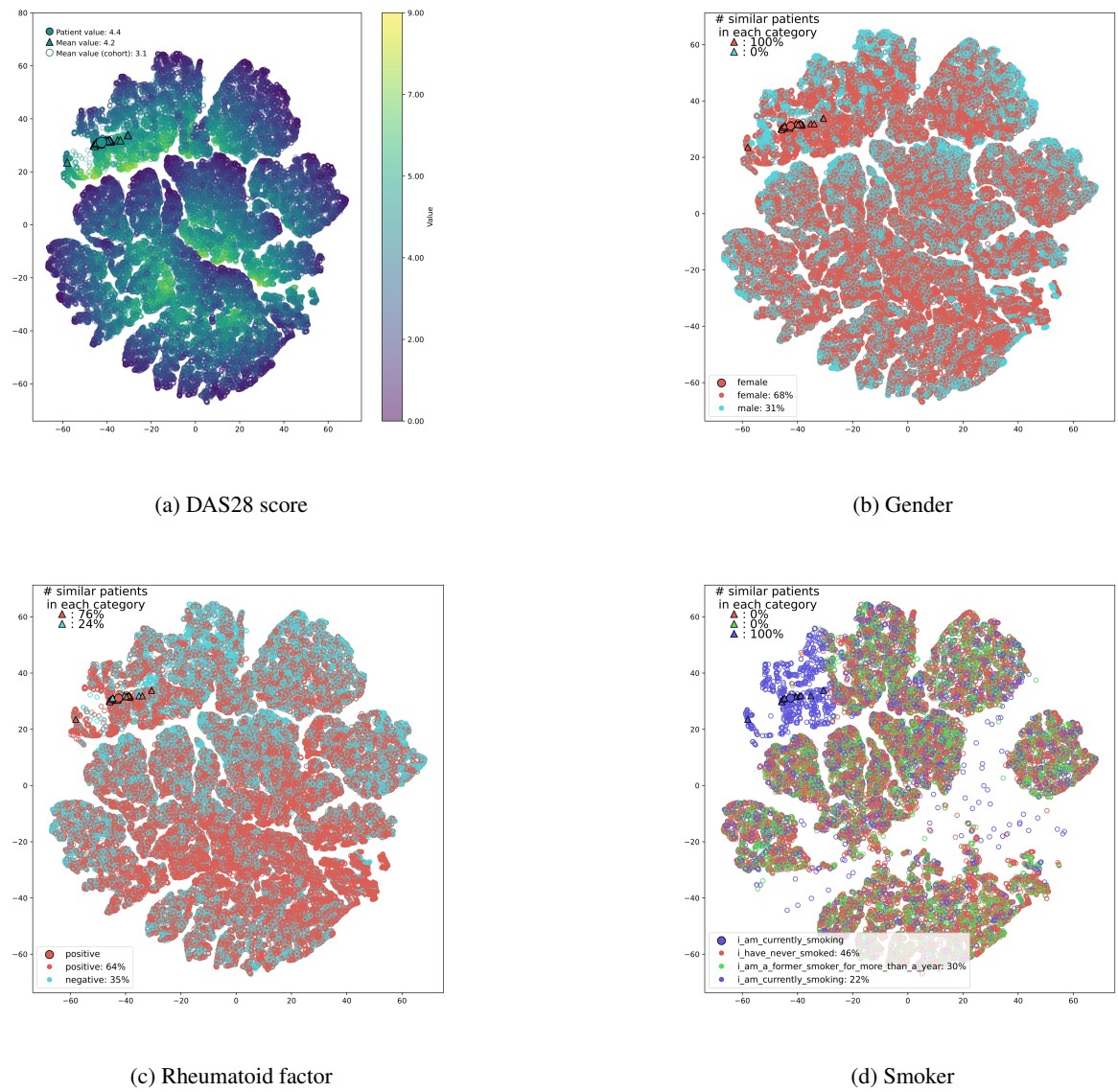

(a) DAS28 score

(b) Gender

(c) Rheumatoid factor

(d) Smoker

Figure 3: **t-SNE visualisation of patient representations.** Each point shows the t-SNE embedding of a representation of a patient at a given time. The subplots show the decomposition overlaid with the feature values (restricted to the embeddings with an available value for the feature). Furthermore, we highlighted a patient from the test set (larger filled dot) and her nearest neighbours (triangles) as computed by our algorithm. For each continuous feature, we compute the average value in the entire cohort and within the subset of nearest neighbours (top left of the plot). For categorical features, we computed the proportion of each category (bottom left of the plot).

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

# A. Patient similarity

## A.1. Feature importance computation

We describe the procedure for computing the empirical feature importance for patient similarity assessment. For continuous features, we computed the average absolute distance (AAD) between the feature value of the patients in the test set ($\mathcal{R}_{test}$) and the average value in their matched set $\mathcal{N}_e$ (in the training data):

$$AAD = \frac{1}{\mid \mathcal{R}_{test} \mid} \sum_{e \in \mathcal{R}_{test}} \mid x_e^c - \frac{1}{\mid \mathcal{N}_e \mid} \sum_{e' \in \mathcal{N}_e} x_{e'}^c \mid,$$

where $x_e^c$ is the value of the continuous feature $c$ for patient embedding $e$. A low AAD for a feature indicates that subsets of similar patients tend to have similar values. For all computations, we restricted the subsets to the embeddings with available feature $c$.

For categorical features, we compared the prior empirical distributions of each category with their probability distributions within the subsets of nearest neighbours. For a categorical feature $X_j$ with possible categories $S_j$, we computed the empirical probability of each category $k \in S_j$ in the train set. We also computed the adjusted probabilities for the embeddings in the neighbourhood $\mathcal{N}_e$ of an embedding $e$ with feature value $k$ from the test set, i.e. the probability $P(x_{e'}^j = k \mid x_e^j = k, e' \in \mathcal{N}_e)$, where $x_e^j$ is the category of feature $j$ for embedding $e$. For an embedding $e' \in \mathcal{R}_{train}$, the prior empirical probability $P(x_{e'}^j = i)$ of category $i \in S_j$ is

$$P(x_{e'}^j = i) = \frac{\sum_{e \in \mathcal{R}_{train}} \mathbb{1}\{x_e^j = i\}}{\sum_{e \in \mathcal{R}_{train}} \sum_{k \in S_j} \mathbb{1}\{x_e^j = k\}},$$

and the adjusted probability is

$$P(x_{e'}^j = k \mid x_e^j = k, e' \in \mathcal{N}_e)) = \frac{\sum_{e \in \mathcal{R}_{test}} \mathbb{1}\{x_e^j = k\} \sum_{e' \in \mathcal{N}_e} \mathbb{1}\{x_{e'}^j = k\}}{\sum_{e \in \mathcal{R}_{test}} \mathbb{1}\{x_e^j = k\} \sum_{e' \in \mathcal{N}_e} \sum_{i \in S_j} \mathbb{1}\{x_{e'}^j = i\}}.$$

The magnitude of the increase in adjusted probability versus prior probability reflects the importance of the feature for the similarity computation.

## A.2. Ranking

| Feature | AAD | Standardised AAD |
|---|---|---|
| asdas_score | 0.25 | 0.24 |
| das283bsr_score | 0.35 | 0.25 |
| n_painfull_joints_28 | 2.05 | 0.41 |
| n_painfull_joints | 2.40 | 0.43 |
| crp | 5.35 | 0.46 |
| n_swollen_joints | 2.14 | 0.50 |
| bsr | 8.03 | 0.50 |
| mda_score | 0.73 | 0.56 |
| n_enthesides | 1.42 | 0.58 |
| joints_type | 8.05 | 0.61 |
| haq_score | 0.46 | 0.65 |
| pain_level_today_radai | 1.91 | 0.71 |
| activity_of_rheumatic_disease_today_radai | 1.89 | 0.71 |
| hb | 0.97 | 0.72 |
| height_cm | 6.73 | 0.73 |
| weight_kg | 12.05 | 0.76 |

Table 2: **Similarity metric: contribution of continuous features.** Average absolute distance (AAD) and standardised AAD between the feature value of a test embedding $e_{p,t}$ and the mean feature value within its nearest neighbours $\mathcal{N}_e$. The features are ordered by standardised AAD. We see that the two DAS and the number of painful joints are taken into account the most during the similarity assessment.

| Category $c$ | Base $P(c)$ | Adjusted $P(c \mid x_e = c)$ | Increase (percentage) |
|---|---|---|---|
| morning_stiffness_duration_radai: 2_to_4_hours | 0.04 | 0.08 | 100.0 |
| morning_stiffness_duration_radai: more_than_4_h... | 0.02 | 0.04 | 100.0 |
| gender: male | 0.29 | 0.46 | 59.0 |
| morning_stiffness_duration_radai: 1_to_2_hours | 0.08 | 0.11 | 38.0 |
| morning_stiffness_duration_radai: all_day | 0.03 | 0.04 | 33.0 |
| smoker: i_am_currently_smoking | 0.23 | 0.27 | 17.0 |
| ra_crit_rheumatoid_factor: negative | 0.37 | 0.43 | 16.0 |
| morning_stiffness_duration_radai: 30_minutes_to... | 0.16 | 0.18 | 12.0 |
| gender: female | 0.71 | 0.78 | 10.0 |
| morning_stiffness_duration_radai: no_morning_st... | 0.47 | 0.51 | 9.0 |
| ra_crit_rheumatoid_factor: positive | 0.63 | 0.68 | 8.0 |
| smoker: i_am_a_former_smoker_for_more_than_a_year | 0.31 | 0.33 | 6.0 |
| anti_ccp: negative | 0.38 | 0.40 | 5.0 |
| anti_ccp: positive | 0.62 | 0.63 | 2.0 |
| smoker: i_have_never_smoked | 0.46 | 0.47 | 2.0 |
| morning_stiffness_duration_radai: less_than_30_... | 0.21 | 0.21 | 0.0 |

Table 3: **Similarity metric: contribution of categorical features.** Empirical probability of a category $c$ versus adjusted probability. The increase in the adjusted probability reflects the importance of a given category in the similarity assessment. Longer durations of morning stiffness and gender have the strongest impact on the similarity assessment.

