# OpenReview forum: "Explainable Deep Learning for Disease Activity Prediction in Chronic Inflammatory Joint Diseases"
_ICML.cc/2023/Workshop/IMLH — IMLH 2023 Poster_

### Official Review · Reviewer_A7n6 · 2023-06-17
**An explainable multi task framework that meets clinical intuition**

**Rating:** 7
**Confidence:** 4

**Review:**

Authors propose an explainable multi-task model to encode patient history into a comparable representation
that can predict future disease activities. The multi task approach of connecting different data sources, assigning an importance to each score makes clinical sense. Developing on the method, can help assist clinical decision making. Feature ranking is another promising avenue.

For an expanded version of the paper, stronger benchmarks should be considered.

---

### Official Review · Program_Chairs · 2023-06-19
**Explainable Deep Learning for Disease Activity Prediction in Chronic Inflammatory Joint Diseases**

**Rating:** 6
**Confidence:** 3

**Review:**

In this study, the authors proposed an explainable attention-based neural network model trained on data from patients with different arthritis subtypes to predict future disease activity scores. The network transforms longitudinal patient journeys into comparable representations, enabling additional case-based explanations through computed patient journey similarities. The results demonstrated the effectiveness of the proposed method, and the paper is easily comprehensible. However, I have several concerns regarding this research.

Firstly, I would like clarification on whether Table 1 presents the results based on the model with both global and local attention or only global attention. If Table 1 displays the results of the former, please also provide the results of the model with global attention only. The reason for this request is that the authors found that the contribution matrix (CM) greatly influences the prediction results based on global attention, and subsequently added local attention weights for the CM in the patient's history when predicting the target outcome (DAS).

Secondly, it would be beneficial if the authors could emphasize the contribution of their work and highlight the differences between the proposed method and the state-of-the-art techniques in the field.

---

### Meta-Review · Area_Chair_a8Di · 2023-06-19

**Recommendation:** Accept (Poster)
**Confidence:** 4

**Metareview:**

Reviewers are generally positive in recommending the acceptance of this manuscript but also raise concerns. Please address them in the final version.

---

### Decision · Program_Chairs · 2023-06-20

Accept (Poster)